

# HTCNN-Attn: a fine-grained hierarchical multi-label deep learning model for disaster emergency information intelligent extraction from social media

Shanshan Li[1,2], Qingjie Liu[1,2] and Xiaoling Sun[1]

[1] School of Information Engineering, Institute of Disaster Prevention, Beijing, China
[2] Hebei Province University Smart Emergency Application Technology Research and Development Center, Sanhe, China

## ABSTRACT

To address the challenge of extracting fine-grained emergency information from noisy social media during disasters, we propose HTCNN-Attn, a hierarchical multi-label deep learning model. It integrates a three-level tree-structured labeling architecture, Transformer-based global feature extraction, convolutional neural network (CNN) layers for local pattern capture, and a hierarchical attention mechanism. The model employs a hierarchical loss function to enforce label consistency across three levels: binary disaster filtering (Level 1), mid-level category classification (Level 2), and fine-grained subcategory extraction (Level 3). Experiments on the Appen and HumAID datasets demonstrate superior performance, achieving an accuracy of 0.9725, a Micro-F1 of 0.7402, and a hierarchical consistency (HC) score of 0.821, outperforming state-of-the-art baselines. Ablation experiments validate the importance of hierarchical modeling, Transformer encoding, CNN layers, pre-trained embeddings, and the hierarchical attention mechanism. Cross-event generalization tests on the CrisisBench dataset demonstrate robust generalization (HC = 0.678), while its lightweight design enables efficient real-time deployment (12.0 ms latency). A case study of the 2015 Nepal earthquake validates its practical utility, where the model accurately classified tweets into hierarchical labels and routed structured information to support emergency response coordination. This demonstrates the effectiveness of the proposed model in supporting rapid, efficient, and fine-grained emergency response after disasters, thereby enhancing disaster response capabilities.

# INTRODUCTION

Natural disasters such as earthquakes, floods, typhoons, and tornadoes are complex, uncertain, and dynamic events. They frequently cause substantial economic losses, casualties, and widespread environmental damage. The inherent unpredictability of these events complicates accurate and timely prediction. Consequently, robust disaster management strategies, which incorporate prevention, mitigation, and rapid response,

Corresponding author
Shanshan Li, liss@cidp.edu.cn

are essential to mitigate their impacts. These strategies depend critically on real-time and accurate data to support situational awareness (SA), defined as the process of gathering and interpreting disaster-related information for effective decision-making and response (*Zhai, 2022*). Traditional SA techniques, such as social surveys, remote sensing, and meteorological observations, often fall short in real-time applications due to time-consuming data collection and limited coverage (*Wu, Wu & Ye, 2020*; *Joyce et al., 2009*). Given these limitations, there is an urgent need for alternative data sources and methods to improve situational awareness in disaster management.

The proliferation of mobile Internet and social media platforms (*e.g.*, Twitter, Facebook, and Weibo) has positioned social media as a critical source of real-time, on-the-ground information during disasters (*Hagras, Hassan & Farag, 2017*; *Wang et al., 2023*). Individuals affected by or witnessing disasters share firsthand information *via* text, images, and videos, making social media an indispensable supplementary data source for situational awareness. Recent studies emphasize the advantages of social media in disaster response, particularly its rapid data transmission and reliability during emergencies when traditional communication channels fail. However, the vast volume of social media posts, often accompanied by noise and irrelevant content, complicates the task of filtering and extracting meaningful information. This challenge is further intensified by the diverse informational needs of different emergency response agencies, each requiring tailored, relevant data for effective response coordination.

Early studies on social media-based disaster information extraction relied heavily on statistical approaches, such as keyword-based filtering. For example, *Olteanu et al. (2014)* developed a keyword dictionary for disaster detection on Weibo, while *Olteanu, Vieweg & Castillo (2015)* used crowdsourcing to categorize crisis information. *Qi, Qi & Su (2020)* applied sentiment analysis to evaluate public emotional responses during the 2017 Jiuzhaigou M7.0 earthquake in Sichuan Province, China. Their approach utilized predefined sentiment dictionaries to quantify emotional fluctuations preliminarily. *Liu et al. (2022)* applied statistical methods to analyze the spatiotemporal features of public sentiment related to the 2018 Yunnan Yangbi earthquake. Additionally, *Han, Wang & Bu (2018)* explored the application of statistical methods for web-based disaster event information extraction. Although these methods provided basic filtering capabilities, they were limited in scalability and accuracy.

The advent of machine learning has driven significant advancements, enabling more automated and precise extraction of disaster-related information. *Imran et al. (2013)* trained machine learning models to first identify disaster-related information in tweets, and then constructed a machine learning model for multi-class classification of informative messages, including categories such as warnings, advice, deaths, injuries, offers of help, missing persons, and general population information. *Ashktorab et al. (2014)* developed "Tweedr," a machine learning tool that applied algorithms such as sLDA, SVM, and logistic regression to identify disaster-relevant tweets, demonstrating improved accuracy. *Pipek, Liu & Kerne (2014)* explored the use of machine learning in crisis informatics and collaboration tools. Their study proposed a collaborative framework based on machine learning for the automatic extraction of crisis-related information from social

media. *Wang & Ruan (2018)* used machine learning techniques, including SVM and Decision Trees, to classify earthquake-related information and analyze spatiotemporal patterns from social media data. *Bo et al. (2018)* used models such as Random Forests and support vector machines (SVMs) to analyze social media information and infer earthquake intensity. However, these methods can only extract shallow semantic features, ignoring the text structure and the semantic information in the context, and are relatively weak in handling long sentences.

With the emergence of deep learning, researchers have leveraged its capability to capture semantic and structural text features, enabling more accurate identification of disaster-related information. *Caragea, Silvescu & Tapia (2016)* developed a binary classifier based on convolutional neural networks (CNN) to identify disaster-related content in tweets, demonstrating significant performance improvement across multiple flood event datasets. *Bhere et al. (2020)* utilized Twitter data from Kaggle to build feature vectors *via* Word2Vec and trained a CNN-based classifier for disaster-related tweet detection. *Madichetty & Muthukumarasamy (2020a)* developed binary classification models using various deep learning techniques, including CNNs, long short-term memory (LSTM) networks, bidirectional LSTM (BLSTM), and BLSTM with attention, to identify disaster-related content. *Karteris et al. (2022)* targeted Twitter, classifying tweets as informative (disaster-related) or non-informative (not disaster-related) using machine learning (ML) techniques to improve situational awareness for first responders. *Maulana & Maharani (2021)* introduced a BERT-MLP model that combines geospatial data to enhance disaster tweet classification accuracy.

Recent advancements in deep learning have led to even more specialized methods for disaster information extraction. *Lin et al.* applied the BERT (Transformer-based pre-trained model) to classify disaster-related information, demonstrating the superiority of BERT's pre-trained ability in disaster classification tasks. *Qi (2022)* proposed a deep learning framework that integrates remote sensing data with social media analytics for earthquake disaster assessment. This approach highlights the potential of deep learning in enhancing disaster risk evaluation. *Jin et al. (2021)* combined CNNs and recurrent neural networks (RNNs) to analyze sentiment dynamics across user groups during typhoon events, revealing intricate emotional patterns. *Vongkusolkit & Huang (2021)* provided a comprehensive review of deep learning-based social media classification, especially in the context of natural disasters, emphasizing its advantages in extracting deep features from unstructured data to improve disaster management. *Madichetty & Muthukumarasamy (2020b)* introduced an enhanced crisis data classification method for Twitter, significantly improving classification performance using contextual representations and deep neural networks. Additionally, *Toraman, Aydin & Kaya (2023)* focused on detecting help-seeking messages during earthquake disasters, emphasizing the importance of real-time situational awareness. *Wu, Wu & Ye (2020)* provided a review on the use of social media data in natural disaster emergency management, highlighting its potential for rapid decision-making. *Zhai (2022)* used deep learning models such as LSTM networks and CNNs as part of their multi-level analytic framework to analyze Twitter data for disaster situational awareness. *Andreadis et al. (2022)* presented the DisasterMM task, which involves

**Table 1 Summary of key literature on situational awareness and disaster response using social media.**

| Author(s) | Methodology | Disaster type | Key approach |
| --- | --- | --- | --- |
| Caragea, Silvescu & Tapia (2016) | Convolutional Neural Networks (CNN) for binary classification of disaster-related content in tweets. | Floods | Convolutional Neural Networks (CNN) |
| Olteanu et al. (2014) | Keyword-based filtering for disaster detection on Weibo. | General (Floods, Earthquakes) | Statistical methods (Keyword Filtering) |
| Olteanu, Vieweg & Castillo (2015) | Crowdsourcing for categorizing crisis information. | General (Crises, Earthquakes) | Crowdsourcing |
| Imran et al. (2013) | Machine learning models for multi-class classification of disaster-related information in tweets. | Earthquakes, Floods, Typhoons | Machine learning (Multi-class Classification) |
| Ashktorab et al. (2014) | Machine learning tool "Tweedr" using algorithms like sLDA, SVM, and logistic regression for disaster tweet identification. | General (Earthquakes, Floods) | Machine learning (sLDA, SVM) |
| Bhere et al. (2020) | Twitter data from Kaggle, Word2Vec for feature vector construction, CNN for binary classification. | General (Floods, etc.) | Word2Vec, CNN |
| Madichetty & Muthukumarasamy (2020b) | Deep learning models (CNN, LSTM, BLSTM, BLSTM with attention) for binary classification of disaster content. | General (Earthquakes, etc.) | CNN, LSTM, BLSTM, Attention LSTM |
| Karteris et al. (2022) | Machine learning (ML) techniques to detect informative posts on Twitter. | General (Disasters in general) | Machine Learning (ML) |
| Maulana & Maharani (2021) | BERT + MLP with geospatial data | Earthquakes, Floods | Introduced a BERT-MLP model using geospatial data for better classification of disaster tweets. |
| Toraman, Aydin & Kaya (2023) | BERT-based supervised classification model for identifying help-seeking messages during earthquakes. | Earthquakes | BERT, Supervised text classification |
| Yin et al. (2024) | Instruction fine-tuned large language model (e.g., GPT-based) for multi-label classification of disaster tweets. | General (Disasters in general) | Large Language Models (LLMs), Multi-label classification |

multimedia analysis of disaster-related social media data, demonstrating the integration of diverse media types in disaster response.

Table 1 summarizes the methods and key findings from the studies, illustrating the evolution of techniques for disaster information extraction. The table categorizes approaches into statistical methods, machine learning, and deep learning models used to classify disaster-related content from social media. This progression from basic statistical methods to advanced deep learning approaches reflects the growing capability to handle the complex, large-scale data associated with social media during disasters.

Present research efforts are largely directed towards two main tasks:

(1) Identification of informative messages: This binary classification task distinguishes disaster-related from non-disaster-related information. Although successful in filtering relevant content, this approach lacks the granularity needed to address the diverse needs of response organizations, such as distinguishing between messages on aid requests and donation needs.

(2) Multi-class and multi-label classification: Given the complexity of disaster scenarios, many studies now tackle multi-class and multi-label classification, aiming to categorize messages across overlapping categories.

A critical challenge lies in developing models capable of hierarchical multi-label classification, as disaster-related messages often span multiple interconnected categories. To address these challenges, we propose a hierarchical situational awareness model with fine-grained categorization for disaster information extraction from social media. The proposed HTCNN-Attn model integrates the Transformer architecture for global context modeling, CNN layers for local pattern extraction, and a hierarchical attention mechanism to achieve precise multi-label classification of disaster information. This model implements a three-level labeling system for disaster-related data, allowing emergency response organizations to quickly access relevant information aligned with their specific roles. Evaluated on the Appen (*Appen, 2024*) and HumAID (*Alam et al., 2024*) disaster response dataset, this approach demonstrates significant performance improvements over conventional models, achieving high accuracy and robustness in disaster classification tasks.

This work makes the following contributions: (1) We design a three-level hierarchical labeling architecture for disaster information extraction, enabling more precise classification of disaster-related content. (2) We propose the HTCNN-Attn model, which combines Transformer and optimized CNN layers with a hierarchical attention mechanism for improved classification performance. Additionally, we design a comprehensive hierarchical loss function to address label dependencies and ensure consistency across levels. (3) We conduct extensive performance testing, an ablation study, and cross-domain generalization experiments demonstrating the superiority of the HTCNN-Attn model over baseline methods across various evaluation metrics. (4) We deploy the HTCNN-Attn model in an emergency management system to enable disaster emergency information intelligent fine-grained hierarchical extraction from social media.

The remainder of this article is structured as follows: "Materials and Methods" describes the materials and methods, including the three-level hierarchical labeling architecture, the disaster perception framework, the HTCNN-Attn model, and model evaluation. "Results" presents the experimental setup, evaluation metrics, results, and deployment, including a case study on the Nepal earthquake. "Discussion" discusses the findings, their applications, limitations, and implications for future research and disaster management. Finally, "Conclusions" summarizes the contributions and outlines directions for future research.

# MATERIALS AND METHODS

## Three-level hierarchical labeling architecture for disaster information

Different levels of emergency response organizations focus on varying levels of granularity in disaster information. Government organizations are more concerned with high-level information, such as determining whether a social media post is disaster-related.

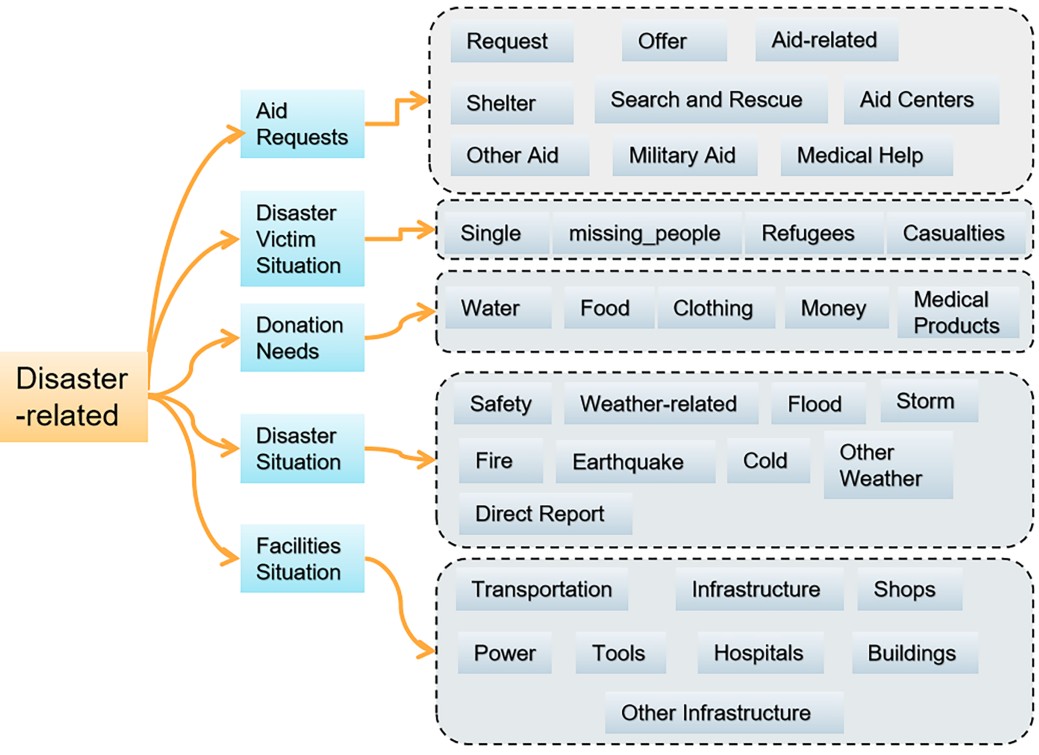

**Figure 1 Three-level hierarchical labeling architecture for disaster information.**

Emergency management departments focus on different categories of disaster awareness, such as aid requests, conditions of disaster victims, and so on. Ground-level emergency management action departments pay more attention to finer-grained categories, such as water, food, safety, transportation, medical assistance, *etc.*

To address these hierarchical needs, we designed a three-level labeling architecture for disaster information on social media, combining the existing categories in the Appen disaster response dataset with emergency management needs, as shown in Fig. 1. The first-level label consists of a binary classification to filter disaster-related information. The second-level labels consist of five categories: aid requests, conditions of disaster victims, donation requests, disaster, and facility conditions. The third-level labels correspond to the second-level labels and further subdivide into nine, four, five, nine and eight categories, totaling 35 labels.

## Disaster perception framework

(1) The disaster perception framework based on social media mainly consists of two processes: the model training process (represented by solid lines) and the real-time data classification and dispatch process (represented by dashed lines), as shown in Fig. 2.

(2) Training data acquisition: Crawling disaster-related tweets from social media platforms such as Twitter or Weibo using web crawlers after a disaster occurs.

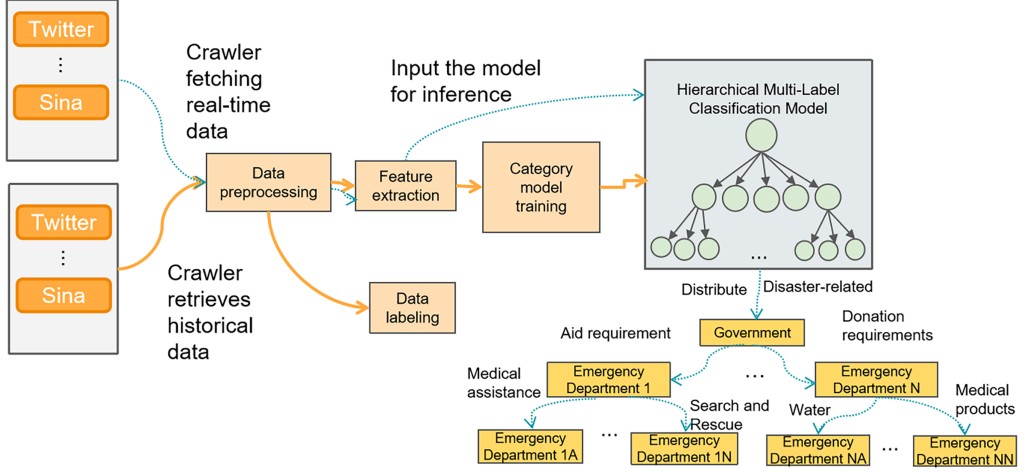

**Figure 2 Disaster perception framework based on social media.**

(3) Training data preprocessing: Removing noise data, handling missing values, and processing duplicate content, *etc*. Text normalization is also carried out, including removing special characters and converting to lowercase.

(4) Training data labeling: The training data is labeled with a three-level tag system for disaster-related information on social media.

(5) Feature extraction for training: The text data is transformed into more informative feature representations.

(6) Training the classification model: The extracted features are input into the model as word embeddings for training a hierarchical multi-label classification model. The model is then evaluated.

(7) Real-time data acquisition: After a disaster occurs, a real-time web crawler is launched to collect relevant tweets.

(8) Real-time data preprocessing: This process is the same as step 2.

(9) Real-time data feature extraction: This process is the same as step 4.

(10) Model inference: The real-time data features are input into the trained model for inference and classification.

(11) Classification result dispatch: Based on the first-level tag classification result, information related to the disaster is filtered out. The second-level classification results are dispatched to the corresponding emergency response departments, and the third-level sub-classification results are sent to the next level of emergency response departments.

## HTCNN-Attn hierarchical multi-label classification model

We have built the HTCNN-Attn hierarchical multi-label classification model, whose architecture is illustrated in Fig. 3. It consists of two main parts: feature extraction and the hierarchical fusion multi-label classification model.

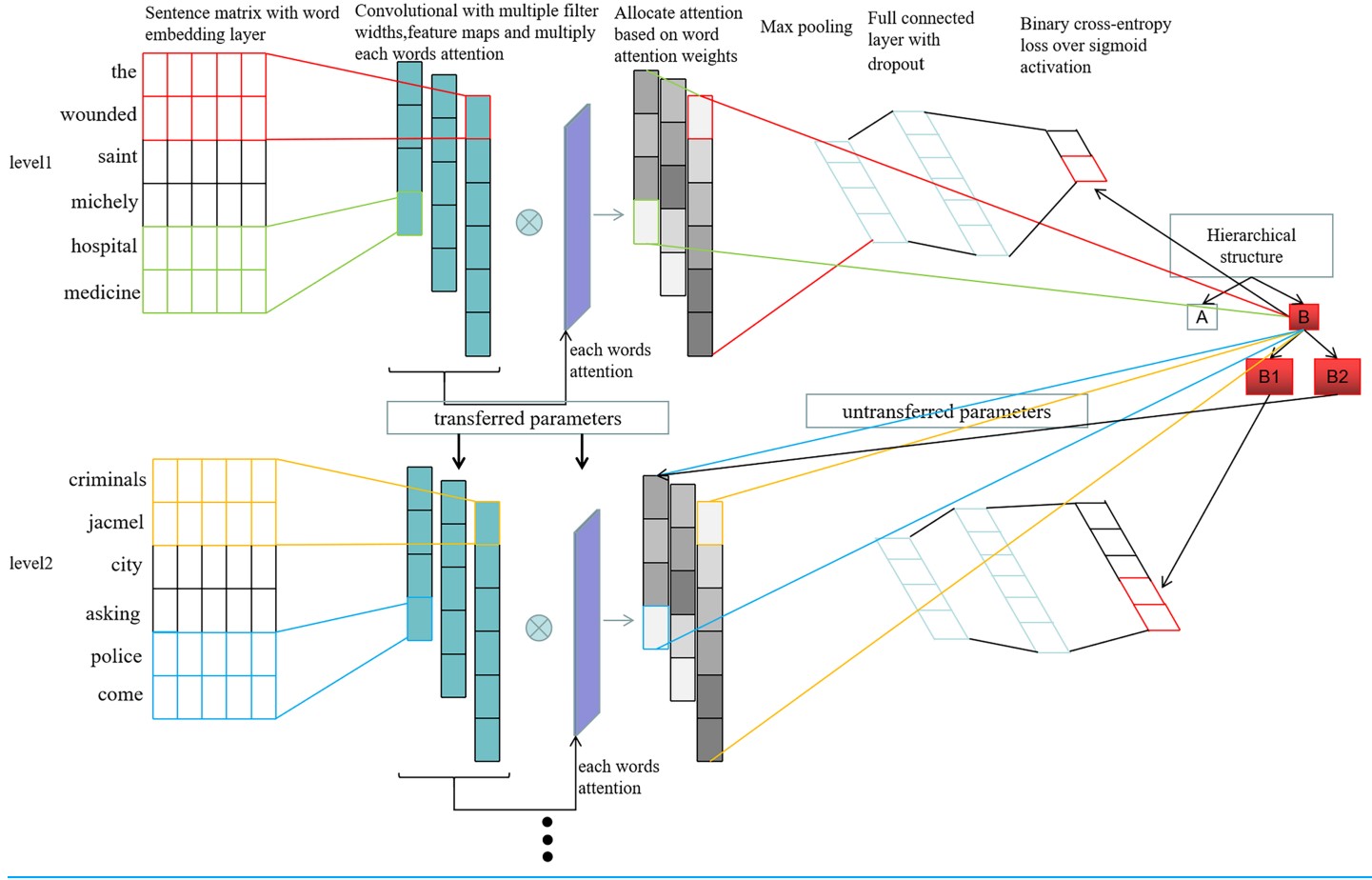

**Figure 3 HTCNN-Attn model network architecture.**

## Feature extraction

Before inputting social media tweets into a deep learning model, it is necessary to first extract meaningful features. Commonly used feature extraction methods include:

Bag of Words (BoW) model: The text is divided into individual words or phrases, and then the frequency of each word in the text is counted to form a vector representation. This method ignores the order and semantic information of the words.

TF-IDF: This method combines the concepts of term frequency (TF) and inverse document frequency (IDF). TF measures the importance of a word in a specific text, while IDF measures the importance of a word in the entire *corpus*. By multiplying TF and IDF, we obtain the TF-IDF value for each word as a feature. However, this method cannot capture word order information, ignores semantic similarity between words, and does not account for document length or the higher weight of frequent words.

N-gram model: The text is divided into sequences of continuous N words. This can be unigrams (single words), bigrams (two consecutive words), trigrams (three consecutive words), *etc*. The N-gram model can capture local relationships between words. However, it

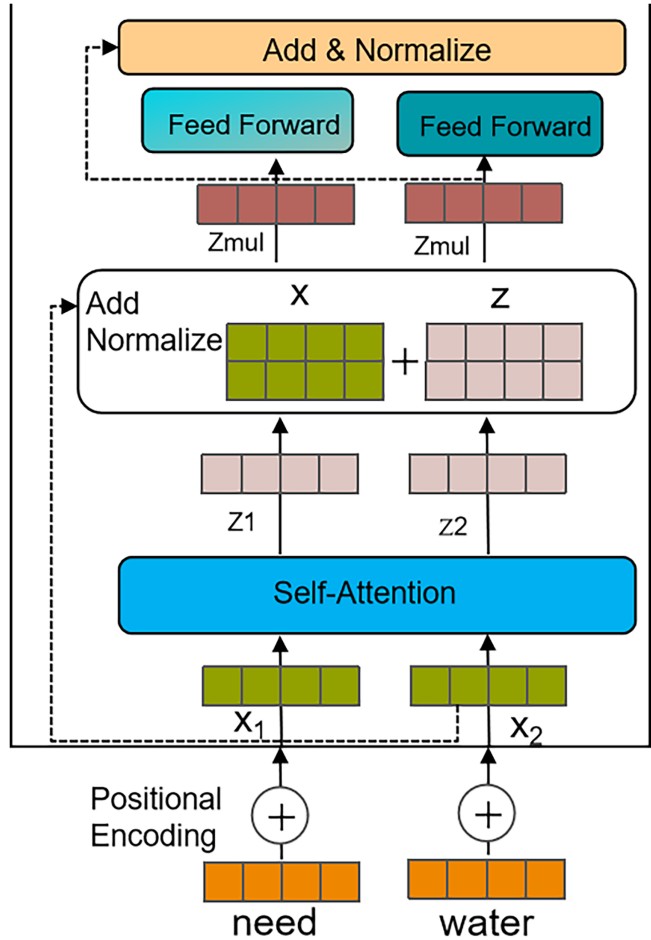

**Figure 4 Feature extraction process based on the Transformer.**

cannot handle long-distance dependencies between words and requires dealing with a large number of sparse features.

Word2Vec model: Based on a neural network, this is a word embedding model that maps each word to a continuous vector space (*Li et al., 2021*). Word2Vec represents semantic relationships between words as distances or similarities between vectors. However, its word vectors are static—*i.e.*, the vector for each word is fixed after training and does not change based on context or tasks. Therefore, it lacks the ability to dynamically adapt to the context of the text. As a result, it is not suitable for handling polysemy or complex semantic relationships between word combinations.

Transformer model (*Vaswani et al., 2017*): The Transformer model, using multi-head self-attention mechanisms and feedforward neural network layers, is highly effective at capturing contextual and semantic information in long texts. It also offers excellent parallelism when handling sequence-based tasks.

Therefore, after a disaster occurs, we use distributed crawlers to collect social media text data and extract features based on the Transformer model. The process is illustrated in Fig. 4, and the specific steps are as follows:

(1) Calculating positional encoding.
In the input sequence's word embedding vectors, positional information is injected to capture the sequential order in the text. The calculation is shown in Eqs. (1) and (2).

$$PE_{2i}(p) = \sin\left(p/10000^{2i/d_{pos}}\right) \tag{1}$$

$$PE_{2i+1}(p) = \cos\left(p/10000^{2i/d_{pos}}\right). \tag{2}$$

(2) Calculating self-attention mechanism.
The text data is quantified into features F, and the output and weight matrices $W^Q$, $W^K$, and $W^V$ are calculated separately to increase the inter-feature relationships and enhance the generalization of the features. The calculation process is shown in Eqs. (3)–(5).

$$Q = FW^Q \tag{3}$$

$$K = FW^K \tag{4}$$

$$V = FW^V. \tag{5}$$

(3) Calculating the output feature Z of the self-attention layer.
The values after softmax normalization are multiplied by the V vector matrix, and the weighted vectors are summed to produce the self-attention output for that position. The calculation process is shown in Eq. (6).

$$Z = \text{Softmax}\left(\frac{QK^T}{\sqrt{d_k}}\right)V \tag{6}$$

where $d_k$ is the dimension of the model.

(4) Calculating the output Matrix $Z_{mul}$ of the multi-head self-attention layer.
Where H denotes the number of attention heads, and $Z_i$ represents the $(i + 1)$-th attention head. $[\cdots]$ represents the concatenation of the H attention heads, and $W^O$ denotes the concatenated matrix. Summing and layer normalization are performed, with layer normalization denoted by LN. The specific process is shown in Eqs. (7) and (8).

$$Z_{mul} = [Z_0, Z_1, \ldots, Z_i]W^O \tag{7}$$

$$Z_{mul} = LN(Z_{mul} + F). \tag{8}$$

(5) Output features.
The Z_mul is passed to the feedforward neural network, where it undergoes a fully connected layer and a nonlinear transformation using an activation function to obtain the final output features. Afterward, summing and layer normalization are applied again to further enhance the representational power and stability of the features.

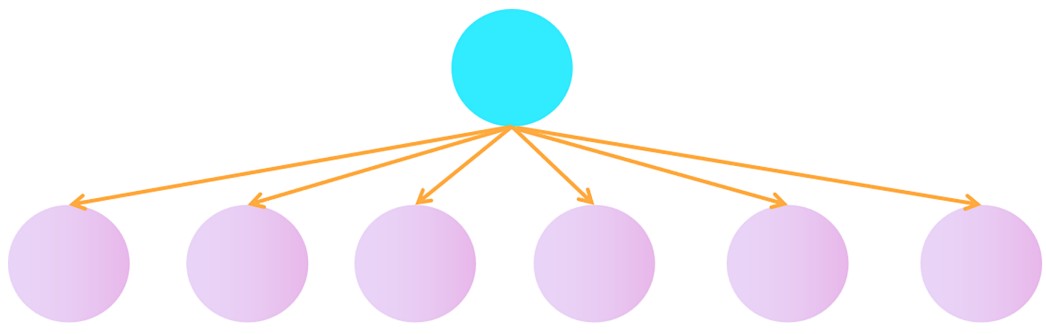

**Figure 5**  **Flat structure of multi-label classification.**

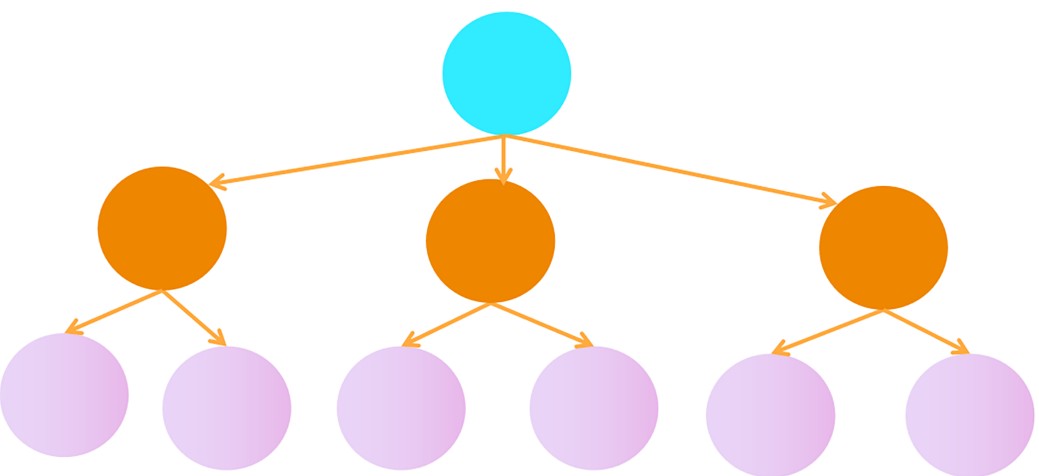

**Figure 6**  **A tree hierarchical structure of multi-label classification.**

### *Hierarchical fusion multi-label classification model*

In traditional multi-label classification problems, each sample can be assigned multiple labels, typically using a flat structure, as shown in Fig. 5. In this problem, we propose a hierarchical multi-label tree structure, as shown in Fig. 6. In this tree structure, higher-level labels represent more general categories, while lower-level labels represent more specific subcategories. There is a hierarchical relationship between the labels.

We designed and implemented a hierarchical multi-label classification model by integrating CNNs to fuse hierarchical information. Figure 7 illustrates the workflow of the proposed model. This model was designed to classify three hierarchical categories (cat_1, cat_2, and cat_3).

First, the input layer (main_input) accepts text sequences as a 1D tensor of shape (max_len,). The extracted feature vectors are then used for word embedding operations. The embedding layer converts integer word indices into dense vector representations. Next, three convolutional layers—cnn1, cnn2, and cnn3—are defined for three different convolutional window sizes (3, 4, 5), followed by max-pooling operations. Convolution operations are performed in all three layers, with the calculation process shown in Eqs. (9)–(11).

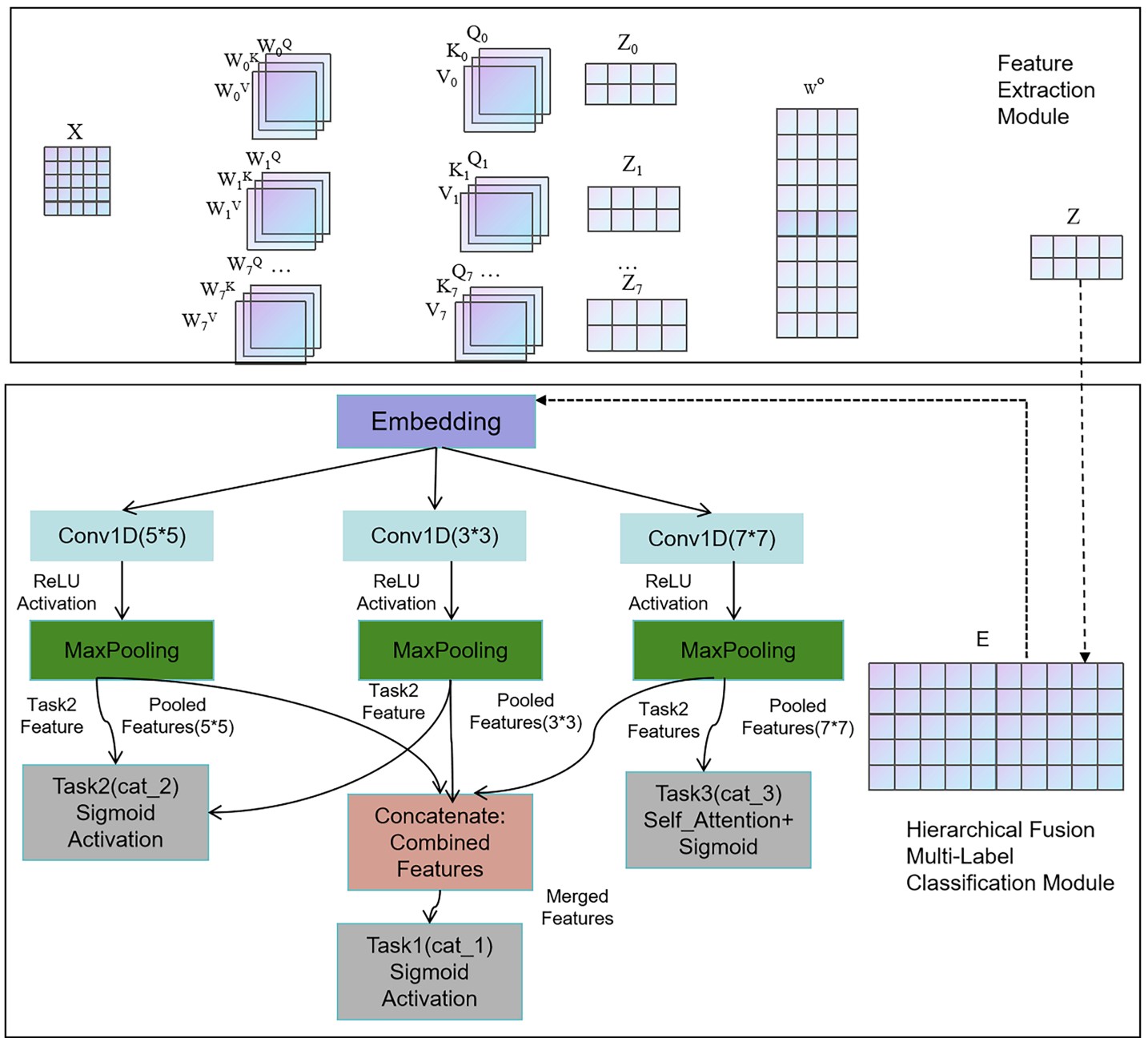

**Figure 7 The fusion flowchart.**

$$r_i = relu(W_{lk} \bullet E_{i:i+h_l-1} + b) \tag{9}$$

$$R = [r_1, r_2, \ldots, r_{n-h_l+1}] \tag{10}$$

$$M_l = [R_1, R_2, \ldots, R_K]. \tag{11}$$

Here, $W_{lk}$ represents the weights of the k convolutional kernels at the l-th layer. $r_i$ represents a new feature of a word vector sequence, b is the bias term, and ReLU is the activation function. After passing through the convolutional kernels, the word vector

features are transformed into a feature map $M_l$ at the l-th layer, which consists of multiple feature vectors.

The attention weights for each word vector in the feature map $M_l$ are computed as follows, with the calculation process described in Eqs. (12)–(15).

$$avg = mean(M_l) \tag{12}$$
$$max = \text{Max}(M_l) \tag{13}$$
$$Attention = sigmoid(W_t \bullet concat(avg, max)) \tag{14}$$
$$M_l = M_l \bullet attention. \tag{15}$$

Here, 'avg' denotes the average across the channel dimension of $M_l$, while 'max' indicates the maximum value along the channel dimension of $M_l$. We concatenate the average and maximum values and then multiply them by a weight matrix with a kernel size of t. After applying the sigmoid activation function, we obtain the attention weights for each word. Next, we perform max pooling on each feature vector $R_i$ of $M_l$, generating a new feature representation $F_l$. After passing through a fully connected layer, we obtain the classification results for layer l. The calculation process is described in Eqs. (16)–(18).

$$\hat{r}_i = \text{max\_pool}(R_i) \tag{16}$$
$$F_l = \left[\hat{r}_1, \hat{r}_2, ..., \hat{r}_K\right] \tag{17}$$
$$F_{out} = linear(F_l). \tag{18}$$

### Hierarchical loss function

In hierarchical multi-label classification tasks, the hierarchical loss function plays a crucial role in ensuring consistency between labels and handling label dependencies. Traditional loss functions may not effectively capture these dependencies, leading to inconsistent predictions between labels. To address this issue, we propose a hierarchical loss function that incorporates weighted loss and hierarchical consistency penalty mechanisms.

(1) Weighted hierarchical loss function

In hierarchical multi-label classification, the importance of each label may vary depending on its hierarchical depth. Typically, lower-level labels have a greater impact on the final decision and should be assigned higher weights. The weighted loss function assigns different weights to each label to reflect the hierarchical dependencies between labels, ensuring that incorrect predictions are penalized more heavily for lower-level labels. The weighted loss function formula is shown in Eq. (19).

$$\text{Loss} = \sum_{i=1}^{n} w_i \cdot \text{BCE}(y_i, \hat{y}_i) \tag{19}$$

where n is the total number of labels, $y_i$ is the true value of the i-th label, $\hat{y}_i$ is the predicted value of the i-th label. The weight $w_i$ is typically inversely proportional to the label's hierarchical depth. Higher-level labels have lower weights, while lower-level labels have higher weights. $\text{BCE}(y_i, \hat{y}_i)$ is the binary cross-entropy loss that calculates the loss for the i-th label.

(2) Hierarchical consistency penalty

The hierarchical consistency penalty mechanism aims to reduce inconsistencies between labels, particularly in hierarchical structures. If a child label is predicted as positive (*i.e.*, 1), but its parent label is predicted as negative (*i.e.*, 0), an additional penalty is applied to maintain consistency across levels. Hierarchical Consistency Loss Formula is shown in Eq. (20).

$$\text{Consistency Loss} = \sum_{i=1}^{n} \sum_{j \in P(i)} \lambda_{ij} \cdot \left| \hat{y}_i - \hat{y}_j \right| \tag{20}$$

where n is the total number of labels. P(i) represents the set of parent labels of label i in the hierarchy. $\hat{y}_i$ is the predicted value of label i. $\hat{y}_j$ is the predicted value of label j (the parent label). $\lambda_{ij}$ is a penalty coefficient that increases the loss if a child label is predicted as 1 but its parent label is predicted as 0.

(3) Combined hierarchical loss function

By combining the weighted loss and the hierarchical consistency penalty, the final hierarchical loss function is shown in Eq. (21).

$$\text{Total Loss} = \text{BCE Loss} + \lambda_{\text{consistency}} \cdot \text{Consistency Loss} \tag{21}$$

where BCE Loss is the standard binary cross-entropy loss used for multi-label classification. Consistency loss is the hierarchical consistency penalty that handles the hierarchical dependencies between labels. $\lambda_{\text{consistency}}$ is a hyperparameter that controls the strength of the hierarchical consistency penalty.

This combined loss function ensures that the model not only accurately predicts the categories of each label but also maintains consistency between labels at different hierarchical levels, improving the model's performance and interpretability.

### Model evaluation

In hierarchical multi-label evaluation, accuracy is a commonly used metric (*Yi, Geng & Bai, 2023*; *Huang & Liu, 2022*), which measures the proportion of samples predicted correctly by the model across the entire dataset. The calculation process is shown in Eq. (22).

$$\text{Accuracy} = \frac{1}{n} \sum_{i=1}^{n} \delta(Y_i', Y_i) \tag{22}$$

where 1 ($Y_i' = Y_i$) equals 1 if the predicted labels $Y_i'$ match the ground truth labels $Y_i$, and 0 otherwise.

Accuracy serves as a basic evaluation metric for hierarchical multi-label tasks. To provide a more comprehensive evaluation of the performance of hierarchical multi-label models, we also consider micro-F1 and macro-F1 scores (*Zhou et al., 2020*), which account for precision and recall at multiple levels. The calculation of micro-F1 involves Eqs. (23)–(25).

$$\text{Precision}_{\text{micro}} = \frac{\sum_{i=1}^{L} \text{TP}}{\sum_{i=1}^{L} \text{TP} + \sum_{i=1}^{L} \text{FP}} \tag{23}$$

$$\text{Recall}_{\text{micro}} = \frac{\sum_{i=1}^{L} \text{TP}}{\sum_{i=1}^{L} \text{TP} + \sum_{i=1}^{L} \text{FN}} \tag{24}$$

$$\text{Micro\_F1} = \frac{2\,\text{Precision}_{\text{micro}}\,\text{Recall}_{\text{micro}}}{\text{Precision}_{\text{micro}} + \text{Recall}_{\text{micro}}} \tag{25}$$

Macro-F1, on the other hand, is computed across the entire dataset by averaging the precision, recall, and F1 score of all levels. The calculation is shown in Eqs. (26)–(28).

$$\text{Precision}_{\text{macro}} = \frac{\sum_{i=1}^{n} \text{Precision}_i}{n} \tag{26}$$

$$\text{Recall}_{\text{macro}} = \frac{\sum_{1}^{n} \text{Recall}_i}{n} \tag{27}$$

$$\text{Macro\_F1} = \frac{2\,\text{Precision}_{\text{macro}}\,\text{Recall}_{\text{macro}}}{\text{Precision}_{\text{macro}} + \text{Recall}_{\text{macro}}} \tag{28}$$

P@K measures the proportion of correctly predicted labels among the top K predictions, emphasizing the quality of the highest-ranked outputs. It is particularly valuable in multi-label classification tasks where the model produces a ranked list of label probabilities. The calculation is presented in Eq. (29).

$$\text{P@K} = \frac{1}{N} \sum_{i=1}^{N} \frac{\sum_{j \in R(i,K)} 1(j \in Y_i)}{n} \tag{29}$$

where N is the number of samples, R(i, k) denotes the top K predicted labels for the i-th sample. $Y_i$ is the set of true labels for the i-th sample. $1(j \in Y_i)$ equals 1 if the predicted label j is in the true label set $Y_i$, and 0 otherwise.

Hierarchical consistency (HC) evaluates whether the predictions adhere to the hierarchical relationships between labels. Specifically, it ensures that child labels are only activated when their corresponding parent labels are also activated. The consistency score is defined in Eq. (30).

$$\text{Hierarchical Consistency} = 1 - \frac{\text{Inconsistent Predictions}}{\text{Total Checks}} \tag{30}$$

where Inconsistent Predictions denotes the number of instances where a parent label is not predicted, but one or more child labels are predicted. Total Checks denotes the total number of parent-child relationships evaluated across the dataset.

# RESULTS

## Experimental data

We utilize a publicly available disaster response dataset from Appen and HumAID. The Appen dataset contains 26,248 labeled messages sent during various historical disasters worldwide, including those in Chile and Pakistan. HumAID consists of 77,196 annotated

**Table 2 Message preprocessing and hierarchization.**

| Message | Three-layer label |
|---|---|
| There's nothing to eat and water, we starving and thirsty | related, related@assistance, related@assistance@request, related@assistance@aid_related, related@assistance@medical_help, related@donation_needs, related@donation_needs@medical_products, related@donation_needs@water, related@donation_needs@food, related@assistance@other_aid, related@infrastructure_and_utilities_damage, related@infrastructure_and_utilities_damage@infrastructure_related, related@infrastructure_and_utilities_damage@buildings, related@infrastructure_and_utilities_damage@other_infrastructure, related@disaster_situation, related@disaster_situation@weather_related, related@disaster_situation@floods, related@disaster_situation@direct_report |

tweets that have been collected during 19 major natural disaster events, including earthquakes, hurricanes, wildfires, and floods, which happened from 2016 to 2019 across different parts of the World. Appen's data is annotated with one or more of 36 categories. HumAID's data is annotated with 11 categories, including caution and advice, displaced people and evacuations, don't know cant judge, infrastructure and utility damage, injured or dead people, missing or found people, not humanitarian, other relevant information, requests or urgent needs, rescue volunteering or donation effort, sympathy and support.

## Data preprocessing

We integrated the Appen and HumAID datasets. After cleaning (removing duplicates and invalid entries), we reorganized labels into a three-level structure. Table 2 illustrates the hierarchization process of a single message.

To address data imbalance, we balanced the dataset while preserving its hierarchical structure, as illustrated in Fig. 8. At the first level, we applied stratified sampling to balance the "Disaster-related" (58%, 27,144 samples) and "Non-disaster" (42%, 19,656 samples) categories, resulting in a total of 46,800 balanced samples without discarding hierarchical relationships. For the second and third levels, we employed a hybrid approach. The Synthetic Minority Over-sampling Technique (SMOTE) was used to generate synthetic samples for underrepresented subcategories, such as "Medical Help" (increased from 200 to 700 samples in the balanced data) and "Other Infrastructure" (increased from 500 to 944 samples). Random undersampling reduced overly dominant subcategories, *e.g.*, "Shelter" (decreased from 2,000 to 820 samples) and "Water" (decreased from 4,500 to 1,180 samples), to avoid skewed model training. During training, a class-weighted loss function was applied, assigning higher weights to rare labels. For example, in the third-level subcategories, labels like "Medical Help" (700 samples) and "Other Weather" (480 samples) received greater weight in loss calculations compared to dominant labels like "Flood" (870 samples) and "Refugees" (1,500 samples).

Then we cleaned the data, which included:

(1) Basic text normalization: removing special characters, punctuation, converting words to lowercase, removing mentions, hashtags, URLs, and replacing contractions. We substituted numbers with words, removed non-ASCII characters, and extra punctuation.

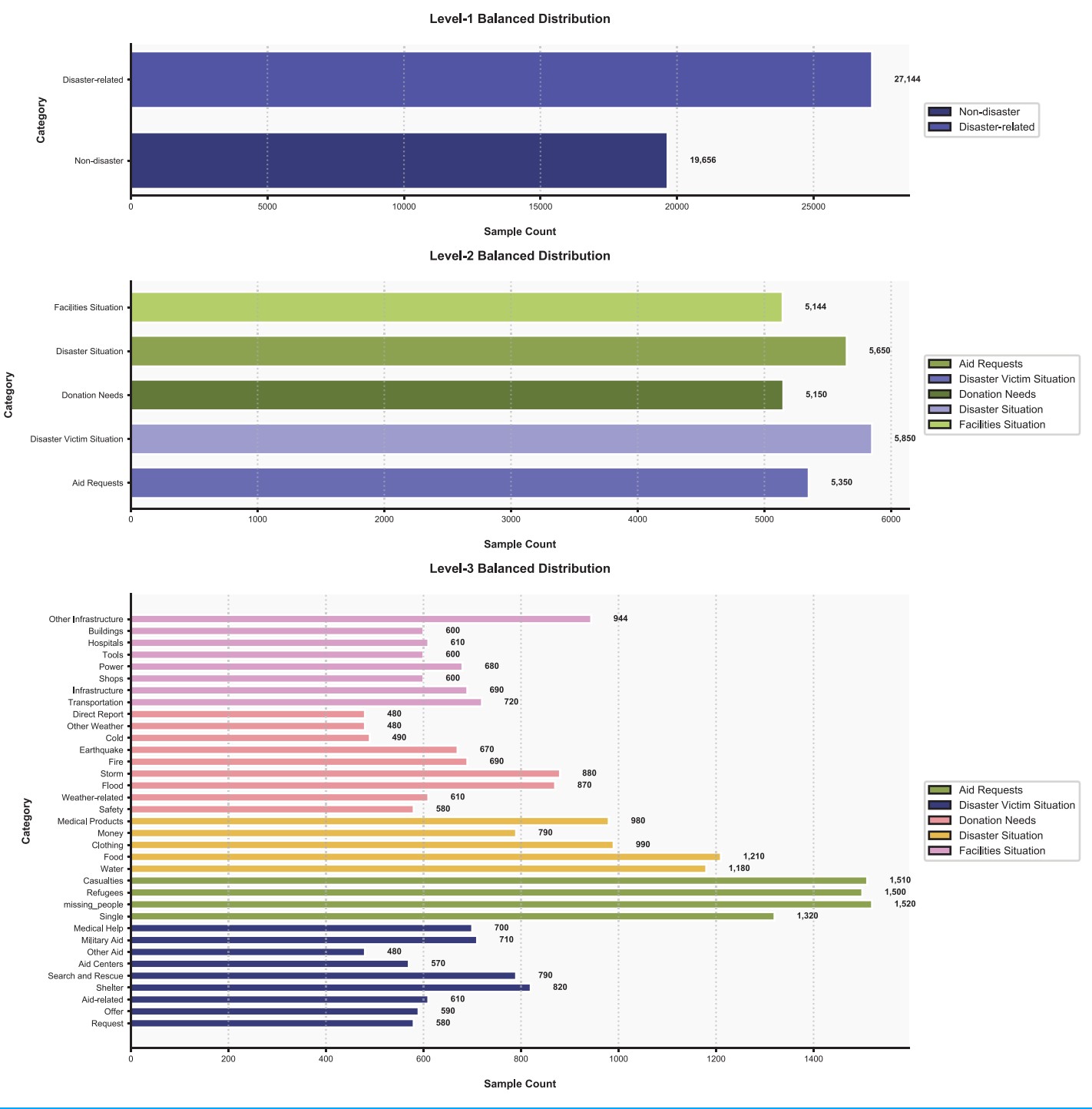

**Figure 8 The distribution of data.**

(2) User identity anonymization: All social media account identifiers (*e.g.*, @usernames, profile URLs) were irreversibly hashed using SHA-256 with dataset-specific salts to prevent re-identification.

(3) Geolocation obfuscation: Precise GPS coordinates were generalized to city-level regions through coordinate truncation (2 decimal places ≈ 1.1 km accuracy) and GeoNames API mapping, ensuring GDPR compliance.

(4) PII redaction: We employed a rule-based NLP pipeline to detect and remove personally identifiable information (PII) such as phone numbers, email addresses, and physical addresses. Additionally, named entities (*e.g.*, individuals' full names) identified *via* spaCy's NER model were masked with generic placeholders (*e.g.*, "[NAME]").

The study utilized only publicly available social media data and adhered to the platform's terms of service. No private or sensitive user content was retained after preprocessing.

## Model training and parameter settings

Experiments were conducted on a Tesla T4 GPU running Linux. The preprocessed data were fed into the HTCNN-Attn model for training under the PyTorch 1.8.1 framework. To ensure optimal performance and reproducibility, we conducted systematic hyperparameter tuning using grid search on a validation subset (10% of the training data). Table 3 summarizes the experimental validation process for critical parameters. For CNN kernels, comparative testing of 128, 256, and 512 units revealed diminishing returns beyond 256 kernels, where the configuration achieved a micro-F1 of 0.740 *vs.* 0.738 with 512 kernels, while reducing training time by 34%. The learning rate was optimized to 2e−6 through stability analysis—this value maximized classification performance (micro-F1: 0.740) while preventing the overshooting observed at 5e−6, which caused a 3.8% micro-F1 degradation. Transformer depth was constrained to four layers after determining that six-layer architectures provided only marginal gains (+0.5% micro-F1) at a 15% increased computational cost. Batch size optimization was settled at 128 to balance GPU memory limitations of the Tesla T4 platform with stable gradient estimation. As shown in Table 3, these parameters collectively maximize classification accuracy while maintaining computational efficiency critical for real-time deployment.

Final parameters were set as: epoch = 50, batch_size = 128, word_embedding_dim = 128, convolution_kernels = 256 (three windows of 3/4/5), learning_rate = 2e−6, dropout = 0.1, activation = Sigmoid, optimizer = Adam, and hierarchical cross-entropy loss.

## Model evaluation

Figure 9 shows the accuracy of the three hierarchical levels over training epochs. From the line plots, we observe that the accuracy at the first level remains relatively stable at the beginning of training, and then gradually increases and stabilizes. The accuracy curve shows a steady upward trend until converging to a relatively high level. This suggests that after sufficient training, the first-level model can gradually learn better feature representations, thereby improving classification accuracy.

The accuracy at the second level increases rapidly during the early training stages and then gradually stabilizes. The accuracy curve shows a rapid rise followed by saturation.

**Table 3 Hyperparameter optimization results.**

| Parameter | Tested values | Micro_F1 | Training time (h) | Selected value |
|---|---|---|---|---|
| CNN kernels | 128/256/512 | 0.721/0.740/0.738 | 3.2/4.5/6.8 | 256 |
| Learning rate | 1e−6/2e−6/5e−6 | 0.728/0.740/0.712 | 4.1/4.5/4.3 | 2e−6 |
| Transformer layers | 2/4/6 | 0.729/0.740/0.735 | 3.9/4.5/5.2 | 4 |
| Batch size | 64/128/256 | 0.731/0.740/0.737 | 3.8/4.5/5.1 | 128 |

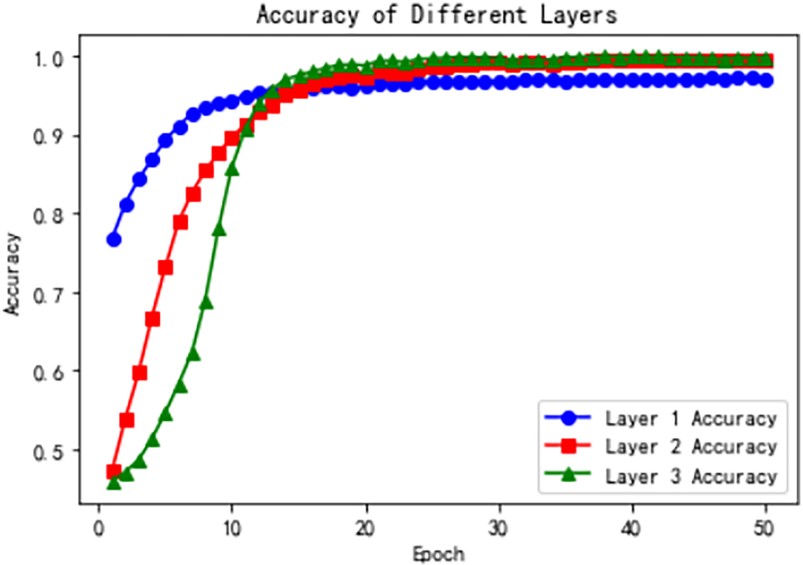

**Figure 9 Model accuracy across hierarchical levels during training.**

This indicates that, building on the first-level features, the second-level model can further extract richer representations and thereby significantly enhance classification accuracy.

The third-level accuracy also shows a gradual increase followed by stabilization. The accuracy curve steadily rises and eventually converges to a relatively high level. This suggests that, inheriting the features from the previous two levels, the third-level model leverages the self-attention mechanism to further enhance feature representation, achieving higher classification accuracy.

Figure 10 illustrates the model loss across the three hierarchical levels and the total loss over training epochs, highlighting the optimization process at each stage. The Level 1 loss starts relatively low and decreases rapidly during the initial epochs, reflecting the simplicity of the binary classification task at this level. By the later epochs, the loss stabilizes near zero, indicating that the model effectively learns high-level representations. At Level 2, the loss starts at a moderately higher value but decreases significantly in the early epochs. It stabilizes around epoch 20, demonstrating the model's ability to leverage Level 1 features for optimizing intermediate classification. The Level 3 loss, dealing with the most fine-grained and complex categories, starts at the highest value and decreases more

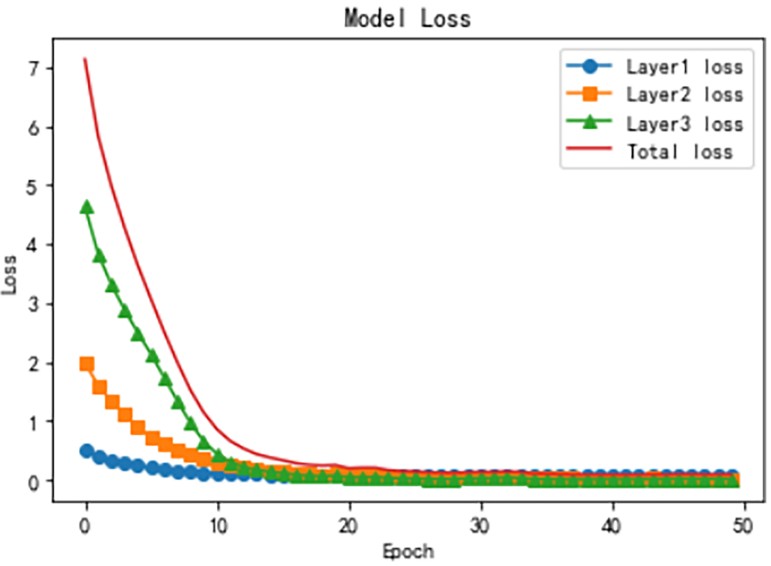

**Figure 10 Training loss dynamics for hierarchical levels.**

**Table 4 Performance and resource efficiency comparison of models.**

| Model | Accuracy | Micro_F1 | Macro_F1 | P@1 | P@3 | HC | Training time (min) | Memory usage (GB) |
|---|---|---|---|---|---|---|---|---|
| TF_IDF-logistic regression | 0.9438 ± 0.021 | 0.6631 ± 0.035 | 0.6587 ± 0.032 | 0.7032 ± 0.040 | 0.6534 ± 0.038 | 0.4523 ± 0.051 | 2.25 ± 0.15 | 0.12 ± 0.01 |
| TF_IDF-Naive Bayes | 0.9317 ± 0.023 | 0.6357 ± 0.038 | 0.6290 ± 0.036 | 0.6745 ± 0.042 | 0.6128 ± 0.040 | 0.4276 ± 0.049 | 1.80 ± 0.10 | 0.10 ± 0.01 |
| ALBERT-BiLSTM | 0.9407 ± 0.019 | 0.6469 ± 0.033 | 0.6643 ± 0.030 | 0.7321 ± 0.036 | 0.6789 ± 0.034 | 0.5564 ± 0.047 | 135.5 ± 8.5 | 0.85 ± 0.04 |
| ALBERT-BiGRU | 0.9513 ± 0.016 | 0.6946 ± 0.028 | 0.6721 ± 0.025 | 0.7643 ± 0.032 | 0.7056 ± 0.030 | 0.6045 ± 0.043 | 150.75 ± 9.0 | 0.92 ± 0.05 |
| stella_en_400M_v5 | 0.9600 ± 0.018 | 0.7215 ± 0.027 | 0.6980 ± 0.024 | 0.7890 ± 0.029 | 0.7320 ± 0.027 | 0.6500 ± 0.041 | 285 ± 12 | 1.62 ± 0.10 |
| HTCNN-Attn | 0.9725 ± 0.015 | 0.7402 ± 0.030 | 0.7001 ± 0.023 | 0.8057 ± 0.025 | 0.7482 ± 0.023 | 0.7241 ± 0.038 | 210 ± 15 | 0.78 ± 0.05 |

gradually, stabilizing after sufficient training, indicating the model's capacity to capture nuanced patterns using hierarchical attention mechanisms. The total loss, aggregating all levels, exhibits a steep decline in the early epochs followed by a gradual stabilization, reflecting the overall convergence and robustness of the HTCNN-Attn model. These trends, combined with the accuracy improvements shown in Fig. 9, demonstrate the model's effectiveness in progressively refining representations and achieving logical consistency across the hierarchical levels.

To validate the HTCNN-Attn model, we compared it with five baselines (TF-IDF-Logistic Regression, TF-IDF-Naive Bayes, ALBERT-BiLSTM, ALBERT-BiGRU, stella_en_400M_v5) using 30 independent trials, reporting metrics as mean ± SD (Table 4). A one-way ANOVA revealed significant overall differences in classification metrics (Accuracy: $F(5, 174) = 28.7$, $p < 0.001$; Micro-F1: $F(5, 174) = 19.5$, $p < 0.001$) and computational efficiency (Training Time/Memory: Kruskal-Wallis $H = 89.3/92.1$, $p < 0.001$). *Post-hoc* Tukey's HSD tests showed HTCNN-Attn significantly outperformed all

baselines in Accuracy (*e.g.*, +1.25% *vs.* stella_en_400M_v5, q(6, 174) = 8.10, *p* < 0.001, Cohen's d = 0.87) and micro-F1 (*e.g.*, +1.87% *vs.* stella_en_400M_v5, q(6, 174) = 6.50, *p* < 0.01), with the largest improvements against traditional models (*e.g.*, +4.38% Accuracy *vs.* TF-IDF-Naive Bayes). For efficiency, it required 210 ± 15 min training time and 0.78 ± 0.05 GB memory, significantly faster and more memory-efficient than deep learning baselines (*e.g.*, −26% time/−52% memory *vs.* stella_en_400M_v5, *p* < 0.001). Shapiro-Wilk and Levene's tests confirmed normality and variance homogeneity (all *p* > 0.05), validating parametric tests. Results collectively demonstrate HTCNN-Attn's robust performance and efficiency in hierarchical disaster information extraction.

### Error analysis

The hierarchical multi-label nature of the task introduces unique challenges for error analysis. Unlike traditional single-label classification, each social media post can be assigned multiple third-level labels (*e.g.*, "Medical Help" and "Medical Products" simultaneously), rendering standard confusion matrices inappropriate. Given the complexity of 35 fine-grained third-level categories, we focus on the top 8 most frequent error pairs (Fig. 11) to ensure clarity while capturing dominant error patterns.

The model exhibited significant confusion between semantically overlapping categories. For instance, "Cold" was frequently misclassified as "Storm" (78 errors, 29.1% rate), often due to contextual ambiguity in posts like "freezing temperatures during the storm". Similarly, "Medical Help" and "Medical Products" showed bidirectional errors (63–78 errors, 23.6% rate) when posts mentioned both "doctors" and "medicines". Rare categories with limited training data, such as "Military Aid" (480 samples), suffered reduced recall (0.55).

To address these issues, we propose (1) refining ambiguous labels (*e.g.*, splitting "Other Aid" into subcategories), (2) injecting domain-specific lexicons (*e.g.*, meteorological terms) to enhance semantic distinction. These adjustments directly target the observed error patterns while maintaining computational efficiency.

### Ablation study

To thoroughly investigate the contribution of each component within our HTCNN-Attn model, we conducted a series of ablation experiments. These experiments involved systematically removing or modifying specific modules to assess their impact on the overall performance. We used the full HTCNN-Attn model as the baseline, which integrates hierarchical information, a Transformer encoder, and a CNN layer for feature refinement. The metrics considered include Accuracy, Micro_F1, Macro_F1, P@1, P@3, and hierarchical consistency. Table 5 presents the results of the ablation study. The experimental process is structured as follows:

(1) No-Hier: We remove the hierarchical structure modeling module, treating the classification as a flat multi-label problem. This tests the importance of leveraging parent-child relationships between labels.

(2) No-Transformer: We eliminate the Transformer encoder and rely solely on CNN (and embeddings), thereby assessing the role of self-attention and global context modeling.

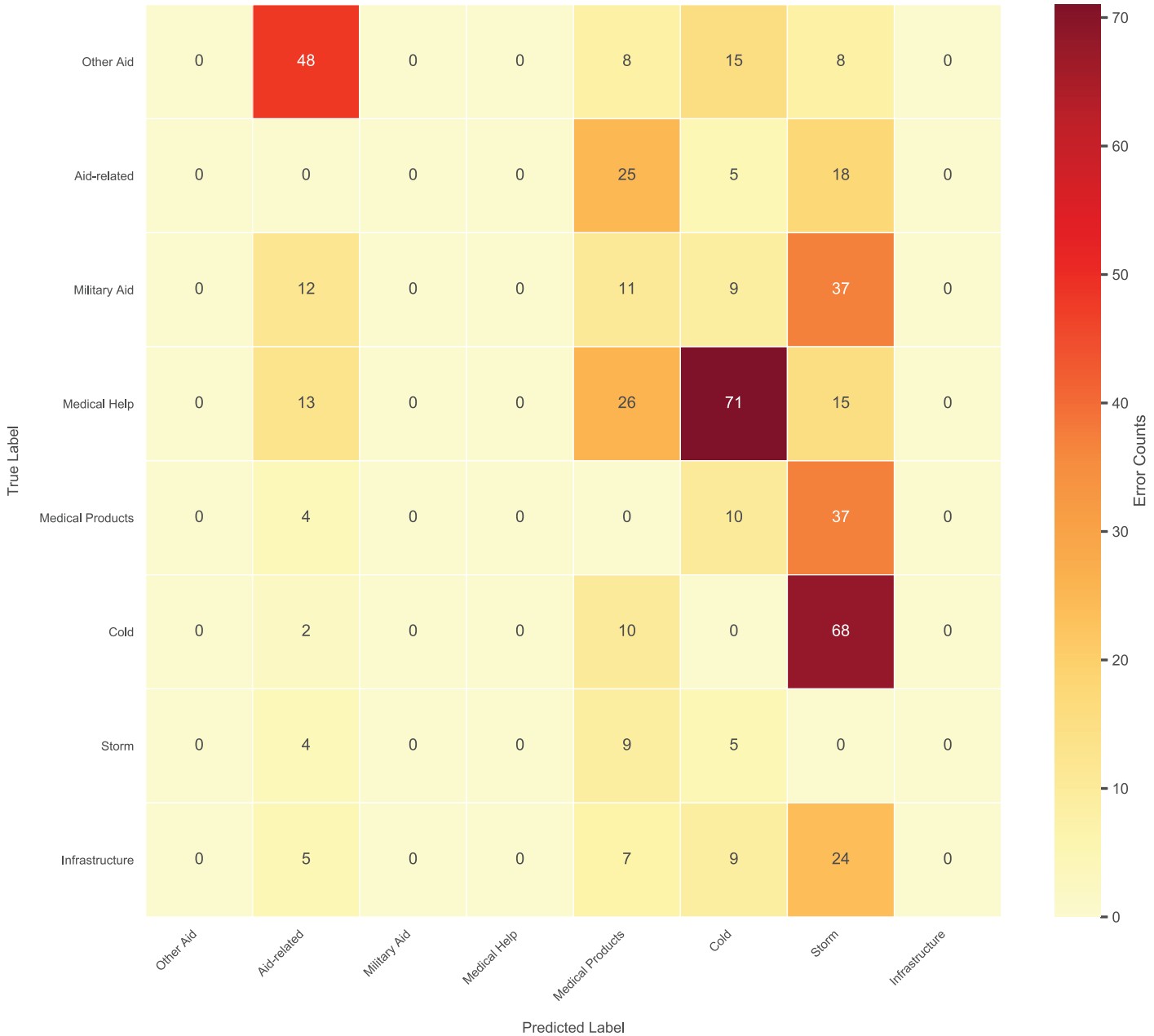

**Figure 11 Top 8 frequent confusion error pairs for third-level labels.**

(3) No-CNN: We remove the CNN layer to determine its contribution to capturing local n-gram patterns and complementing the Transformer's global attention.

(4) No-Pretrained-Embeddings: We replace ALBERT or other pre-trained embeddings with randomly initialized embeddings to gauge the value of external language knowledge.

(5) No-Attention-for-Hierarchy: we remove it to understand how it affects the alignment between label hierarchies and model predictions.

**Table 5  Ablation study results of the HTCNN-Attn model.**

| Model Configuration | Accuracy | Micro_F1 | Macro_F1 | P@1 | P@3 | HC |
|---|---|---|---|---|---|---|
| HTCNN-Attn (Full) | 0.9725 | 0.7402 | 0.7001 | 0.8057 | 0.7482 | 0.7241 |
| No-Hier | 0.9640 | 0.7253 | 0.6904 | 0.7805 | 0.7306 | 0.6207 |
| No-Transformer | 0.9550 | 0.7121 | 0.6802 | 0.7603 | 0.7104 | 0.6505 |
| No-CNN | 0.9670 | 0.7307 | 0.6958 | 0.7909 | 0.7401 | 0.7002 |
| No-Pretrained-Embeddings | 0.9500 | 0.7003 | 0.6704 | 0.7505 | 0.7006 | 0.6807 |
| No-Attention-for-Hierarchy | 0.9700 | 0.7359 | 0.6981 | 0.8002 | 0.7453 | 0.7104 |

The study shows that the hierarchical structure is crucial for maintaining consistent predictions across different levels of the label taxonomy. When this module is removed, the hierarchical consistency drops by approximately 14.36%. Removing the Transformer encoder leads to a decrease of about 3.82% in micro-F1 and 2.87% in Macro-F1. Taking away the CNN layer causes P@1 and P@3 to drop by about 1.87% and 1.07%, respectively. Discarding pre-trained embeddings leads to a significant decline in all metrics, with hierarchical consistency dropping by about 6.06%. Removing the hierarchical attention mechanism results in a 1.9% decrease in hierarchical consistency. Each component of the HTCNN-Attn model contributes significantly to its overall effectiveness, jointly enabling the model to perform better.

## Generalization experiments

To assess the cross-event generalization ability of HTCNN-Attn, we conducted experiments on the CrisisBench dataset (*Firoj Alam et al., 2021*), focusing on the 2012 Hurricane Sandy, a disaster event not included in our original training dataset, HumAID. This choice allows us to rigorously test the model's adaptability to unseen scenarios.

The data processing steps followed the methods described in "Preprocessing", with additional normalization for event-specific language features. We fine-tuned HTCNN-Attn using an AdamW optimizer with an initial learning rate of 0.0001 for 20 epochs. A stratified 70/15/15 split was applied across five independent trials to ensure balanced data representation.

As shown in Table 6, HTCNN-Attn outperformed baseline models, achieving an accuracy of 0.935, a micro-F1 score of 0.702, and an HC score of 0.678. Compared to ALBERT-BiGRU, it improved micro-F1 by 4.2% and HC by 5.5% (both $p < 0.05$). Although the micro-F1 score decreased by 3.7% relative to HumAID, the stable HC score demonstrates the model's robust hierarchical reasoning ability across different disaster events, validating its generalization performance.

## Deployment and inference

Taking the Nepal earthquake that occurred on April 25, 2015, at 14:11 as an example, we collected post-earthquake tweets from Twitter. We utilized the Flask framework to build a system that loads the trained model. By inputting a tweet into the system, accurate classification was achieved. Furthermore, by integrating the location information of the

**Table 6 Model performance on CrisisBench (Hurricane Sandy 2012).**

| Model | Accuracy | Micro_F1 | Macro_F1 | P@1 | P@3 | HC |
|---|---|---|---|---|---|---|
| TF_IDF-logistic regression | 0.8925 | 0.6012 | 0.5985 | 0.6442 | 0.5835 | 0.4118 |
| TF_IDF-Naive Bayes | 0.8818 | 0.5757 | 0.5712 | 0.6228 | 0.5616 | 0.3923 |
| ALBERT-BiLSTM | 0.9007 | 0.5919 | 0.6130 | 0.6815 | 0.6323 | 0.5146 |
| ALBERT-BiGRU | 0.9105 | 0.6438 | 0.6312 | 0.7121 | 0.6634 | 0.5621 |
| stella_en_400M_v5 | 0.9210 | 0.6723 | 0.6580 | 0.7410 | 0.6910 | 0.6120 |
| HTCNN-Attn | 0.9350 | 0.7020 | 0.6881 | 0.7657 | 0.7282 | 0.6780 |

**Figure 12 Hierarchical fine-grained disaster information perception.**

**Table 7 Inference performance comparison of models.**

| Model | Single-sample latency (ms) | Throughput (Samples/Sec) | Memory usage (GB) | FLOPs (GFLOPs) |
|---|---|---|---|---|
| TF_IDF-logistic regression | 8.5 ± 0.3 | 118 | 0.3 | 0.2 |
| TF_IDF-Naive Bayes | 7.2 ± 0.2 | 139 | 0.2 | 0.1 |
| ALBERT-BiLSTM | 19.2 ± 1.1 | 52 | 5.1 | 2.8 |
| ALBERT-BiGRU | 17.0 ± 0.8 | 59 | 4.8 | 2.5 |
| stella_en_400M_v5 | 22.5 ± 1.5 | 44 | 6.3 | 3.6 |
| HTCNN-Attn | 12.0 ± 0.5 | 83 | 3.5 | 1.8 |

Note:
Inference tests were conducted on an NVIDIA Tesla T4 GPU (16 GB VRAM) using PyTorch 2.0.1 and CUDA 11.8. Batch size was set to 1 for single-sample latency measurement. Standard deviations (±) represent results from 100 repeated trials.

tweet with the extracted category information, we were able to enhance the perception of disaster-related information, as illustrated in Fig. 12.

To address real-time deployment requirements, we compared model inference efficiency (Table 7). Traditional Term Frequency-Inverse Document Frequency (TF-IDF) baselines are fast (7.2–8.5 ms latency, 118–139 samples/sec) but lack hierarchical text modeling, limiting accuracy for complex disaster texts. Sequence models like ALBERT-BiGRU offer moderate speed (17.0 ms, 59 samples/sec) but require high GPU memory (4.8 GB), challenging edge use. The large MTEB baseline stella_en_400M_v5, despite strong semantics, is inefficient (22.5 ms latency, 6.3 GB memory), unsuitable for resource-constrained environments. Our

HTCNN-Attn balances speed and efficiency with 12.0 ms latency (29% faster than ALBERT-BiGRU), 83 samples/sec throughput (meeting disaster data peaks), and low memory (3.5 GB, 44% less than stella_en_400M_v5). These metrics validate its readiness for real-time emergency response, proving superior efficiency while maintaining hierarchical classification accuracy.

## DISCUSSION

The experimental results validate the effectiveness of integrating hierarchical information into multi-label classification models, showing significant improvements in predictive accuracy and logical consistency. By incorporating hierarchical cues, the proposed approach aligns with existing research while achieving notable enhancements in label coherence. It effectively addresses challenges such as predicting subcategories without their parent categories, ensuring that higher-level labels influence the classification of lower-level labels. Notably, experiments on the CrisisBench dataset validate HTCNN-Attn's hierarchical architecture in multi-label disaster classification, confirming its robustness and generalizability across unseen real-world scenarios.

However, the study has three key limitations that need attention. First, its reliance on pre-trained embeddings and large labeled datasets (*e.g.*, Appen, HumAID) not only restricts applicability in low-resource or multilingual environments but also leaves its performance untested on non-English platforms like Weibo, and in zero-shot scenarios. Future research could explore lightweight model architectures or transfer learning strategies to reduce dependency on large datasets, and further investigate how to adapt these strategies for non-English data. Additionally, multilingual NLP techniques should be employed to improve cross-lingual generalization, as demonstrated in resource-constrained scenarios. This could include integrating zero-shot learning frameworks to handle unseen disaster types with minimal prior data, expanding the model's utility in novel crisis scenarios.

Second, while the preprocessing pipeline includes basic privacy measures like user anonymization and PII redaction ("Preprocessing"), it lacks integration of advanced techniques such as federated learning (*Kairouz et al., 2021*) and differential privacy (*Dwork, 2008*). These methods are proven to mitigate data leakage risks during distributed model training, as demonstrated in recent crisis informatics research (*Sutedi et al., 2021*; *Bharambe et al., 2024*). Future research should focus on two key directions. First, develop hybrid architectures that combine the hierarchical attention model with edge-based federated learning. This would enable on-device feature extraction, preserving user privacy, critical for emergency response platforms handling multi-jurisdictional social media data. Second, enhance the rule-based PII redaction process with dynamic differential privacy mechanisms to quantify and bound privacy risks when releasing datasets. Additionally, integrating local differential privacy (LDP) during geolocation obfuscation would provide mathematical guarantees against re-identification while maintaining spatial utility for disaster mapping. These advancements balance analytical effectiveness with rigorous privacy protection, ensuring the framework is both robust and ethically compliant.

Third, although this study focuses on lightweight hierarchical modeling for edge deployment, it does not explore the integration of MTEB models' generic embeddings with explicit hierarchical architectures (*e.g.*, hierarchical embedding spaces). Future research may bridge this gap by leveraging federated learning to adapt these embeddings to diverse emergency datasets, thereby balancing generic semantic representation with task-specific label structure modeling—an extension that builds directly on this study's foundational framework.

## CONCLUSIONS

This study presents HTCNN-Attn, a hierarchical deep learning model that enables fine-grained emergency information extraction from social media, addressing challenges of noise, scalability, and hierarchical label dependencies. By developing a three-level, tree-structured labeling system and proposing the HTCNN-Attn model, this research makes significant advancements in hierarchical multi-label classification. The model integrates Transformer-based feature extraction with CNN layers and a hierarchical attention mechanism. Moreover, the hierarchical loss function, which includes both weighted loss and consistency penalties, ensures the model maintains label consistency across different levels of hierarchy, enabling accurate and semantically consistent categorization of disaster-related data.

Experimental evaluations on the Appen and HumAID datasets demonstrate its superior performance, achieving state-of-the-art results in accuracy, micro-F1, and hierarchical consistency. Its lightweight architecture achieves 25% faster training and lower memory usage than state-of-the-art baselines, enabling efficient real-time deployment with a latency of 12.0 ms. Cross-event validation on the CrisisBench dataset confirms its robust adaptability, maintaining strong hierarchical reasoning and outperforming baselines in tasks involving unseen disaster scenarios. In a 2015 Nepal earthquake case study, the model's deployment and inference process effectively routed fine-grained emergency information to support situational awareness for response teams.

Future work should expand multilingual support, integrate federated learning for privacy, and enhance zero-shot capability through meta-learning or cross-domain adaptation. These steps will strengthen scalability for low-resource scenarios and ethical compliance, advancing context-aware disaster response systems.

### Funding

This research was funded by the Langfang Science and Technology Research and Development Program (grant number 2024011014) and the National Key Research and Development Program of China (grant number 2024YFC3908000). The funders had no role in study design, data collection and analysis, decision to publish, or preparation of the manuscript.

## Grant Disclosures

The following grant information was disclosed by the authors:
Langfang Science and Technology Research and Development Program: 2024011014.
National Key Research and Development Program of China: 2024YFC3908000.

## Competing Interests

Shanshan Li, Qingjie Liu, and Xiaoling Sun are employees of Institute of Disaster Prevention. The authors declare that they have no competing interests.

## Author Contributions

- Shanshan Li conceived and designed the experiments, analyzed the data, prepared figures and/or tables, authored or reviewed drafts of the article, and approved the final draft.
- Qingjie Liu conceived and designed the experiments, performed the computation work, authored or reviewed drafts of the article, and approved the final draft.
- Xiaoling Sun performed the experiments, analyzed the data, performed the computation work, prepared figures and/or tables, and approved the final draft.

## Data Availability

The APPEN data is available at GitHub: https://github.com/tri-bui/disaster-response/tree/master/data.

The HumAID data is available at: https://crisisnlp.qcri.org/humaid_dataset.

The CrisisBench data is available at: https://crisisnlp.qcri.org/crisis_datasets_benchmarks.

## Supplemental Information

Supplemental information for this article can be found online at http://dx.doi.org/10.7717/peerj-cs.2992#supplemental-information.

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
