# Peer review of "HTCNN-Attn: a fine-grained hierarchical multi-label deep learning model for disaster emergency information intelligent extraction from social media"

_PeerJ Computer Science, doi:10.7717/peerj-cs.2992_

## Round 0.1 · original submission · Major Revisions

This work has value, but also issues. Please carefully consider the comments and revise the article accordingly.

**Language Note:** The review process has identified that the English language must be improved. PeerJ can provide language editing services - please contact us at [email protected] for pricing (be sure to provide your manuscript number and title). Alternatively, you should make your own arrangements to improve the language quality and provide details in your response letter. – PeerJ Staff

Reviewer 1 ·

Basic reporting

The paper is well-structured and adheres to academic standards, with clear sections covering introduction, methodology, experiments, and conclusions. The literature review effectively contextualizes the challenges of disaster-related information extraction from social media. However, the discussion lacks references to privacy-preserving techniques (e.g., differential privacy, federated learning, or anonymization methods) commonly applied to social media data in disaster scenarios. This omission weakens the ethical grounding of the data collection and preprocessing steps. Additionally, while the experimental datasets (Appen and HumAID) are publicly available, no details are provided about privacy safeguards during data preprocessing, such as user ID anonymization, geolocation obfuscation, or text redaction to remove personally identifiable information (PII). Clarifying these steps is critical to ensure compliance with privacy regulations (e.g., GDPR) and ethical research practices.

Figures and tables are generally well-presented, but verification is needed to confirm no inappropriate image manipulation or redundant visualizations.

Experimental design

The proposed HTCNN-Attn model innovatively integrates Transformer, CNN, and hierarchical attention mechanisms for fine-grained multi-label classification. The three-level labeling architecture addresses a clear research gap in hierarchical dependency modeling. Experimental parameters (e.g., batch size, learning rate) are sufficiently detailed for reproducibility. However, the rationale for hyperparameter selection (e.g., number of CNN kernels, Transformer layers) is not explicitly justified. While data balancing is mentioned (46,800 samples), the specific techniques (e.g., oversampling, class-weighted loss) require clarification. The case study on Nepal earthquake tweets demonstrates practical utility but lacks quantitative metrics on inference speed or computational efficiency, which are essential for real-time deployment validation.

Validity of the findings

Results show significant improvements over baselines (e.g., TF-IDF-Logistic, ALBERT-BiGRU) in accuracy, Micro-F1, and hierarchical consistency. Ablation studies convincingly validate the contributions of hierarchical modeling and attention mechanisms. However, statistical significance tests (e.g., t-tests, ANOVA) are absent, reducing the robustness of performance claims. Generalizability is also limited to Appen and HumAID datasets; cross-validation on other disaster datasets (e.g., CrisisNLP) would strengthen the conclusions. Error analysis (e.g., confusion matrices for misclassified third-level labels) is missing, which could reveal model limitations in fine-grained categorization.

Additional comments

Test model performance on non-English platforms (e.g., Weibo) or with limited labeled data.

Report training/inference time, memory usage, and hardware requirements.

Explicitly address data anonymization protocols and adherence to platform-specific privacy policies.


Reviewer 2 ·

Basic reporting

Generally, the manuscript is well-written and mostly professional in tone. The literature review is comprehensive, with over 30 relevant citations across traditional, machine learning, and deep learning methods. Figures and tables are clear and support the narrative. The manuscript conforms to PeerJ standards in format and structure. The concern is about the baseline chosen. Please refer to the following comments for details.

Experimental design

1 The paper mentions downsampling to 46,800 entries, but doesn't report class distributions post-processing. Please report the class balance at each hierarchical level.
2 While the Nepal Earthquake case study is a good testbed, there’s no indication of how well the model generalizes to unseen disasters or social media domains. A cross-event or zero-shot setting would strengthen the work.

Validity of the findings

The authors are advised to add much stronger baselines. The chosen baseline is, to some extent, weak. Embedding models in the MTEB leaderboard is strongly suggested to be added to comprehensively evaluate the proposed model.

Additional comments

Furthermore, there are some typos in the manuscript, the authors are advised to revise them to ensure proofreading.

---

## Round 0.2 · accepted · Accept

This version satisfied the reviewers successfully. It can be accepted currently. Congrats!

Reviewer 1 ·

Basic reporting

I have no more comments.

Experimental design

I have no more comments.

Validity of the findings

I have no more comments.

Additional comments

I have no more comments.

Reviewer 2 ·

Basic reporting

I am satisfied with the current revision. The authors did fully address my concerns, especially about adding more experiments on baseline models from MTEB.

Experimental design

Excellent after revision

Validity of the findings

Excellent after revision

Additional comments

They have fully addressed my concerns, and this work is strongly suggested to be accepted, considering its contribution to the disaster management field, its model design, extensive experiments, and strong baseline models.